# Chronic Methylmercury Intoxication Induces Systemic Inflammation, Behavioral, and Hippocampal Amino Acid Changes in C57BL6J Adult Mice

**DOI:** 10.3390/ijms232213837

**Published:** 2022-11-10

**Authors:** Tyciane S. Nascimento, Daniel V. Pinto, Ronaldo P. Dias, Ramon S. Raposo, Paulo Iury G. Nunes, Cássia R. Roque, Flávia A. Santos, Geanne M. Andrade, José Lucas Viana, Anne H. Fostier, Alessandra Sussulini, Jacqueline I. Alvarez-Leite, Carlos Fontes-Ribeiro, João O. Malva, Reinaldo B. Oriá

**Affiliations:** 1Neuroscience and Behavior Laboratory, Drug Research and Development Center, Federal University of Ceará, Fortaleza 60430-275, Brazil; 2Laboratory of Tissue Healing, Ontogeny, and Nutrition, Department of Morphology and Institute of Biomedicine, School of Medicine, Federal University of Ceará, Fortaleza 60430-270, Brazil; 3Experimental Biology Core, Health Sciences Center, University of Fortaleza, Fortaleza 60812-020, Brazil; 4Natural Products Laboratory, Department of Physiology and Pharmacology, Federal University of Ceará, Fortaleza 60430-270, Brazil; 5Department of Analytical Chemistry, Institute of Chemistry, University of Campinas—UNICAMP, Campinas 13083-862, Brazil; 6Laboratory of Atherosclerosis and Nutritional Biochemistry, Federal University of Minas Gerais, Belo Horizonte 31270-901, Brazil; 7Coimbra Institute for Clinical and Biomedical Research (iCBR), Institute of Pharmacology and Experimental Therapeutics and Center for Innovative Biomedicine and Biotechnology (CIBB), Faculty of Medicine, University of Coimbra, 3000-548 Coimbra, Portugal

**Keywords:** methylmercury, metabolism, hippocampus, neurotransmission, oxidative stress, neuroinflammation, behavior

## Abstract

Methylmercury (MeHg) is highly toxic to the human brain. Although much is known about MeHg neurotoxic effects, less is known about how chronic MeHg affects hippocampal amino acids and other neurochemical markers in adult mice. In this study, we evaluated the MeHg effects on systemic lipids and inflammation, hippocampal oxidative stress, amino acid levels, neuroinflammation, and behavior in adult male mice. Challenged mice received MeHg in drinking water (2 mg/L) for 30 days. We assessed weight gain, total plasma cholesterol (TC), triglycerides (TG), endotoxin, and TNF levels. Hippocampal myeloperoxidase (MPO), malondialdehyde (MDA), acetylcholinesterase (AChE), amino acid levels, and cytokine transcripts were evaluated. Mice underwent open field, object recognition, Y, and Barnes maze tests. MeHg-intoxicated mice had higher weight gain and increased the TG and TC plasma levels. Elevated circulating TNF and LPS confirmed systemic inflammation. Higher levels of MPO and MDA and a reduction in IL-4 transcripts were found in the hippocampus. MeHg-intoxication led to increased GABA and glycine, reduced hippocampal taurine levels, delayed acquisition in the Barnes maze, and poor locomotor activity. No significant changes were found in AChE activity and object recognition. Altogether, our findings highlight chronic MeHg-induced effects that may have long-term mental health consequences in prolonged exposed human populations.

## 1. Introduction

Mercury (Hg) is a ubiquitous heavy metal occurring in nature, but its environmental levels may be markedly increased due to anthropogenic activities such as illegal artisanal gold mining and industrial wasting [1].

Methylmercury (MeHg) is a highly neurotoxic Hg compound, mainly generated by methylating bacteria in aquatic ecosystems, and may bioaccumulate and biomagnify in the food chain with top-food chain predatory fish showing high MeHg levels [2]. In endemic areas of Hg contamination due to illegal gold mining, often seen in the Amazonian region, riverside populations may be prolongedly exposed [3]. Oral exposure to this neurotoxicant is the major route of MeHg intoxication in riverside populations due to daily fishing and water consumption for family subsistence [3]. Hair Hg analysis has been used to assess the level of intoxication of riverine populations across the Amazon, indicating chronic exposure [4].

MeHg can readily cross the blood–brain barrier (BBB) by the amino acid transporter 1 (LAT1), markedly expressed in astrocytes. Astrocytes may then release MeHg into the extracellular vicinity of the brain, which may be taken by neurons and other glial cells, inducing neuroinflammation and cell apoptosis [5].

MeHg neurotoxicity varies according to the exposure levels, leading to neurological symptoms that may include motor, visual, auditory, and mental impairments [6]. However, chronic MeHg exposure effects on human populations are still not completely understood, especially regarding brain biochemistry, which may lead to psychiatric disorders.

Orally ingested MeHg in mice simulates the form of contamination to which humans are often subjected. Few studies have focused on the impact of MeHg on adult hippocampal amino acids and the potential impact on behavior including memory assessment. Therefore, this work aimed to evaluate chronic MeHg effects on molecular biomarkers of systemic and hippocampal inflammation, oxidative stress, and amino acid levels and their impact on working, spatial, recognition, and learning memory in adult mice.

This study focused on the hippocampus, a known brain area affected by Alzheimer’s disease, as MeHg may contribute to an augmented risk of dementia in vulnerable populations [7].

## 2. Results

### 2.1. MeHg Intoxication and Metabolic Changes

MeHg-intoxicated mice showed a significant increase in the hair Hg levels when compared to the unchallenged control group (MeHg: 27.4 ± 7.92 mg/kg; Control: 0.48 ± 0.15 mg/kg, *p* < 0.001) (Figure 1A). Chronic MeHg intoxication significantly increased weight gain in mice on days 4, 7, 10, 13, 16, and 22 compared to the unintoxicated group. In addition, MeHg increased the delta weight, suggesting a MeHg-obesogenic effect (*p* < 0.05) (Figure 1B,C). MeHg intake significantly increased the plasma levels of the total cholesterol (by 64.48%) and triacylglycerols (by 156.8%) (*p* < 0.05) when compared to the unchallenged control group (Figure 1D,E).

### 2.2. Systemic Inflammation Biomarkers

MeHg-intoxicated mice showed significantly higher plasma levels of TNF (MeHg: 2585 ± 13.85 pg/µg; Control: 1595 ± 19.93 pg/µg, *p* < 0.005) (Figure 2A) and circulating LPS (MeHg: 0.28 ± 0.05 pg/µg; Control: 0.075 ± 0.10 pg/µg, *p* = 0.01) when compared to the unchallenged control group (Figure 2B).

### 2.3. Hippocampal Neuroinflammatory-Related Biomarkers

MeHg-intoxicated mice showed lower IL-4 mRNA transcripts compared to the unchallenged control mice (*p* = 0.004) (Figure 3A). No statistically significant difference was found in the transcriptional levels of all the other studied cytokines (Figure 3B–D). MeHg-challenged mice showed higher MPO (MeHg: 667 ± 0.207 pg/µg; Control: 4.21 ± 0.089 pg/µg, *p* < 0.05) (Figure 3E) and higher MDA levels when compared to the control mice (MeHg: 1.37 ± 0.13 µmol/mg of tissue; Control: 0.91 ± 0.07 µmol/mg of tissue, *p* = 0.008), indicating a significant increase in cell lipid peroxidation (Figure 3F).

### 2.4. Hippocampal Neurochemical Changes

Chronic MeHg intoxication induced higher levels of GABA (MeHg: 14,005 ± 3554; Control: 3402 ± 859.7, *p* = 0.01) (Figure 4A) and glycine (MeHg: 3437 ± 1884; Control: 421.1 ± 149.1, *p* < 0.0001) (Figure 4B) and a significantly lower level of taurine (MeHg: 419.4 ± 71.74; Control: 958.8 ± 149.3, *p* = 0.003) when compared to the unintoxicated control group (Figure 4C). No statistically significant difference was found for tyrosine, glutamate, and BDNF levels between the groups (Figure 4D–F). No significant difference was observed in AChE activity in the hippocampus between groups (Figure 4G).

### 2.5. Exploratory and Locomotor Behavior

MeHg-intoxicated mice showed a lower total distance traveled (*p* = 0.02) (Figure 5A), average speed (*p* = 0.02) (Figure 5B), and the number of rearing (*p* = 0.01) (Figure 5C). In addition, the intoxicated mice showed significantly more grooming behavior (*p* = 0.01) (Figure 5D) when compared to the unchallenged control group.

### 2.6. Memory Assessment

MeHg-challenged mice switched arms similarly in the Y-maze compared to the unchallenged control group, suggesting that they had no deficit in the working memory (Figure 6A). No significant difference was observed in the object recognition index of MeHg-challenged mice compared to unintoxicated controls, suggesting intact recognition memory (Figure 6B).

MeHg-intoxicated mice showed a lower time in the escape tunnel area during the probe test in the Barnes maze (Figure 6C). In addition, they showed a higher latency to enter the escape tunnel in the first and second acquisition tests (Figure 6D), suggesting an impairment in spatial memory acquisition compared to the unchallenged controls.

## 3. Discussion

Our study addressed the effects of chronic MeHg exposure (in water intake for 30 days) on systemic inflammation, hippocampal functional biomarkers, and behavior in adult C57BL6/J mice.

In our protocol, experimental mice received MeHg in drinking water to mimic the most common route of intoxication in humans (oral route). After oral intake, MeHg bioavailability relies on gastrointestinal tract absorption and elimination through feces [5], [8,9]. In our study, the estimated consumption of 0.23 mg of MeHg/kg/day was much lower than that reported by Moreira and colleagues, who estimated that young mice, receiving 40 mg/L for 21 days in drinking water, consumed 6 mg of MeHg/kg/day [10].

Over the 30 days of challenge, a pro-obesogenic effect of MeHg was seen. This finding may be related to MeHg influence on hormones regulating the overall metabolism. Chronic exposure to MeHg may induce body weight gain due to ghrelin-related hypothalamic AMP-activated kinase (AMPK)/uncoupled protein 2 (UCP2) signaling in male mice. Ghrelin is a known orexic gastric hormone that increases appetite [11]. MeHg-induced weight gain in our study reached a plateau in the experiment, suggesting other metabolic adjustments. Another endocrine effect of Hg that may be associated with more weight gain is the dysregulation of thyroid hormones. Elevation in the plasma free thyroxine/triiodothyronine ratio (free T4:T3 ratio) was found in workers who inhaled vapor contaminated with MeHg particles [12]. An elevated T4:T3 ratio is related to increased body weight as well as an increased risk of coronary heart disease [13]. MeHg effects on weight may also be influenced by the intestinal microbiota and environmental conditions [14,15]. More studies are needed to dissect the fine regulation of body growth, fat deposits, and weight gain due to MeHg intoxication in adult mice.

Leocadio and cols (2020) showed that MeHg orally intoxicated-10-week-old C57BL6/J mice (20 mg/L in drinking water for 14 days) showed increased systolic and diastolic blood pressure and total plasma cholesterol levels. MeHg showed an inhibitory action on paraoxonase (PON1). PON1 negatively regulates LDL oxidation with a favored effect in reducing oxidized LDL (LDLox) accumulation in atherosclerotic plaques [14]. It has been shown that 3 mg of MeHg/kg via gavage to young rats for 14 days could reduce PON1 activity, with increased plasma concentrations of LDLox, intercellular adhesion molecule 1 (ICAM1), and monocyte chemoattractant protein 1 (MCP1) [16]. In addition, MeHg induces Hg accumulation in the adult mice liver, leading to hepatic dysfunction with increased TBARS [17] and serum levels of transaminases [6], effects that may partly explain the dyslipidemia seen in our study.

Lipopolysaccharide (LPS), derived from the cell wall of anaerobic Gram-negative bacteria, can reach the bloodstream in small concentrations, even under homeostasis, and may be markedly increased with disturbances in the intestinal epithelial barrier. Of note, MeHg can change the intestinal microbiota composition, affecting the Firmicutes/Bacteroidetes ratio [18]. This finding was also seen with the use of antibiotics, immunosuppressive agents and due to chronic intestinal diseases, positively relating to obesity [19,20]. The rise in the intestinal population of Bacteroidetes may be associated with the increase in circulating LPS induced by Hg intoxication [21].

Neutrophilia occurs in response to damage to the intestinal barrier and increased circulating LPS. LPS-activated neutrophils increased the MPO activity in their azurophilic granules [22]. The action of MPO is mediated by binding with H_2_O_2_ (MPO-H_2_O_2_), forming an oxidative compound. MPO-H_2_O_2_ induces the formation of a potent oxidizing agent, hypochlorous acid (HOCl), which may disrupt the structure of lipids and proteins, amplifying the inflammation cascade [23].

Interestingly, one of the elements regulating the oxidative action of HOCl is taurine. In adult guinea pigs, concomitantly administered taurine (300 mg/kg) and LPS (4 mg/kg) were in a single ip dose. Taurine reduced the concentration of MPO and NO_2_ in the peripheral blood neutrophils compared to the group that received LPS only [22]. Taurine displays an anti-inflammatory effect, diminishing HOCl levels by reacting and forming the taurine compound chloramine (TauCl). After neutrophil apoptosis, TauCl is absorbed by macrophages, is less toxic to the cell than HOCl, and acts as a nuclear signal of the nuclear factor erythroid 2-related factor 2 (Nrf2)-Keap1 pathway [24]. The nuclear translocation of Nrf2 is responsible for activating genes related to the transcription of antioxidant enzymes, reducing cellular apoptosis, H_2_O_2_ levels, and increasing the GSH concentrations in macrophages [25]. Another role of taurine as a regulator of oxidative activity is its interaction with complex I of the mitochondrial electron chain [26].

Peripheral injection of a single dose of LPS (10 mg/kg ip) in eight-week-old rats induced BBB permeability, with an increase in β-amyloid in the hippocampus and an increase in the plasma IL-1β, IL-6, and TNF levels [27].

The increase in MPO levels and higher lipid peroxidation (MDA) due to intoxication by MeHg found in this study may be a consequence of systemic inflammation with an increase in circulating LPS. The possible deleterious effect of MeHg on the intestinal barrier may induce a high rate of bacterial lumen–blood translocation [28].

Recently, it was documented that activated microglia can synthesize and release MPO in the brain tissue [29]. Aslankoc et al. (2018), when injecting a single dose of LPS (5 mg/kg ip) in 12-month-old Wistar rats, found increased neutrophil infiltration and augmented IL-6 and caspase-3 expression in the whole brain [30]. HOCl derived from MPO in brain tissue is associated with BBB impairment and eNOS activation [31]. Circulating levels of MPO can induce increased BBB permeability, affecting the neurovascular unit, with changes in endothelial cell tight junctions [32,33]. We currently do not know whether MeHg-induced endotoxemia in our study can alter the BBB with neutrophilic infiltration and microglial activation. More studies are needed to confirm the contribution of microglia and neutrophils in increasing MPO concentrations in the hippocampus in chronic MeHg intoxication.

Microglia–astrocyte–neuron communication can occur through increases in IL-6 levels in the hippocampus. This IL-6-dependent communication pathway was found in cell cultures challenged with MeHg under concentrations below 1 µM, with microglia–astrocyte crosstalk, leading to the activation of astrocytic P2Y1 receptors by ATP, with IL-6 release in the medium [34]. This pathway could have a neuroprotective compensatory action to reduce MPO levels in the hippocampus, but the increase in IL-6 transcriptional activity in the hippocampus after MeHg challenge was not identified in our study.

Although the IL-6-mediated signaling pathway did not appear to have been altered in the MeHg-challenged animals, a reduction in IL-4 gene expression was seen in the hippocampus. IL-4 has been linked to the modulation of LTP and inflammation. Th2 cell-derived IL-4 is a cytokine with primarily anti-inflammatory effects. The entry of IL-4 into brain tissue can occur by diffusion through astrocytic endfeet, after its release by Th2 cells into the cerebrospinal fluid [12].

Zhao and colleagues challenged three-month-old C57BL6/J mice with a dose of LPS (500 µg/kg and 750 µg/kg ip) for seven consecutive days and found an increase in the plasma and brain tissue concentrations of IL-1β, TNF, and a reduction in IL-4 and IL-10, depicting a scenario of systemic inflammation and neuroinflammation. In addition, an increase in the number of immunoassayed β-amyloid-1-42-positive cells was found. In the presence of IL-4, cultured microglia showed an improved clearance of β-amyloid, suggesting a neuroprotective effect [35]. Thus, the transcriptional reduction of IL-4 in the hippocampus in our study, after the MeHg challenge in adult mice, may indicate impairment of the neuroprotective capacity in MPO-related neuroinflammation. IL-4 induces the macrophage phenotype shift to M2, with a decrease in local inflammation. Immunocompromised BALB/c SCID mice that received bone marrow-derived M2 cells showed a reduced latency time in the acquisition phase, increased time in the target quadrant (probe), and reduced latency five days after the administration time in the reversal tests of the Morris maze [36].

Our study showed that MeHg intoxication affected the cognitive and motor capacity of exposed mice, showing deficits in locomotor behavior in the open field test and a delay in the acquisition of spatial memory observed in the performance for the latency to enter the escape tunnel in the first and second acquisition and the time in the escape tunnel area in the Barnes maze test. Previous studies have shown that low doses of Hg (MeHg at 0.04 mg kg^−1^ day^−1^ for 60 days) are able to induce motor, learning, and memory impairment due to neurochemical dysfunction, neuron and astrocyte cell density loss in the CA1, CA3, hilus, and dentate gyrus regions, with increased levels of MDO and nitrite and decreasing antioxidant capacity against peroxyl radicals in Wistar rats [37,38,39]. Exposure to MeHg also reduced the number of reduced Purkinje cells and the volume of DNA in the cerebellum, indicating a total reduction in the number of cells [40]; these findings and the neurochemical changes may explain the observed motor deficits.

In our study, MeHg intoxication reduced GABA concentrations in the hippocampus. MeHg, even in low doses, was capable of blocking GABAA and GABAB receptors in hippocampus slices. With the blockade of the receptors, the bioavailability of GABA in the synaptic cleft increased, but without altering the glutamatergic neurotransmission [41,42,43]. The effect of MeHg on GABA A receptors is likely related to the alkylation of thiol groups of cysteine residues in the receptor, modifying its conformation, and thus altering Cl^−^ flux and neuronal membrane polarization [41]. Impairment in GABAergic signaling can affect memory and neuroplasticity, with actions in the neurogenesis process from neural stem cells (NSC). GABAB receptors are related to the inhibition of new neurons formation, increasing the expression of Hes5+ NSC [43]. In turn, GABAA receptors are related to the reduction in cell proliferation [44]. GABAB receptors are preferentially found on axon terminals, regulating cell excitability through the modulation of Ca^2+^ and K^+^ channels [45].

Studies involving the impact of MeHg on glycine levels or the expression of its receptors are not conclusive. The sparse current information suggests that Hg, in various forms, reduces the bioavailability of glycine in fish [46]. However, when analyzing cerebellum slices from mice exposed to increasing doses of MeHg (10, 20, and 50 µM), increased glycine concentrations were noticed [47]. Mice exposed for 20 days to 5 mg of MeHg/kg/day showed elevated glycine concentrations in the cerebellum of ataxic animals and the synaptosomal fraction of the medulla in non-ataxic animals [47,48]. The increased concentrations of GABA and glycine found in MeHg-challenged mice suggest the inhibitory regulation of LTP and then related-cognitive impairment.

Exposure to MeHg induced a decrease in taurine levels in the present study. Fluctuations in taurine levels and transporters were associated with cognitive impairments [49]. Many studies have shown the protective effect of taurine supplementation against dementia in animal models, by improving memory impairment, inflammation, and apoptosis [24,50,51,52,53]. Taurine also plays a critical role in hippocampus maintenance in nutrition, neuronal cell growth, and differentiation [54].

In our work, chronic MeHg intoxication in adult mice caused systemic inflammation and endotoxemia, with increased circulating MPO levels. Neutrophil-derived MPO release can increase BBB permeability [26] and cause microglia activation. Activated microglia can release MPO into the hippocampus, further enhancing the neuroinflammatory process. We speculate that the reduction in endogenous taurine concentrations in the hippocampus of challenged animals may have contributed to impaired MPO-derived HOCl and favored oxidative stress. More studies are needed to confirm these findings. How changes in hippocampal inhibitory amino acid levels affect behavior remains to be better understood. Furthermore, how these long-term effects may determine the early onset of neurodegenerative diseases needs further investigation.

We acknowledge that the fur Hg values of the experimental mice were higher than the hair Hg levels in human populations. However, environmental disasters are increasingly frequent worldwide, which can lead to severe and prolonged exposure of Hg to water reservoirs and ocean shores [55] and may have a lasting effect on human health.

This study had some limitations including a more detailed analysis of neurotransmitter signaling and receptor expression and the morphological investigation of immunoprotective signaling pathways such as IL-4, in the hippocampus.

To our knowledge, this is the first study to address the systemic and neurotoxic effects of MeHg in adult mice with changes in the hippocampal amino acid levels. Adult neurogenesis and BBB biomarker involvement in MeHg intoxication need further investigation, along with the suggested neuroprotective mechanism of taurine in the hippocampus. These studies may play a significant role in understanding the long-term impact of MeHg intoxication in riverine populations chronically exposed to this organometal and thus contribute to public health policies with the planning of preventive measures and new treatment strategies.

## 4. Materials and Methods

### 4.1. Experimental Mice

A total of one hundred and twenty C57BL6J male mice (n = 60 intoxicated and n = 60 non-intoxicated) were used in this study. Experimental mice were obtained from the production vivarium of the Experimental Biology Core of the University of Fortaleza (UNIFOR). The 3-month-old adult mice (25–30 g) were randomly assigned to a MeHg-challenged and an unchallenged control group.

Experimental mice were kept in microisolators (five mice/microisolator) with rigorously controlled temperature and air humidity. Animals were exposed to a 12 h/12 h light cycle and had free access to water and food. Mice were weighed every 3–4 days. A weight gain curve (% initial weight) and a delta weight gain (final weight minus initial weight) were measured to address a possible pro-obesity effect.

All study protocols were approved by the University of Fortaleza Institutional Animal Care and Use Committee guidelines and followed the current Brazilian regulations for animal protection.

### 4.2. MeHg Intoxication

MeHg-intoxicated animals were isolated from the controls in a separate room to avoid cross-contamination. Mice were intoxicated by receiving a MeHg solution in drinking water (2 mg/L) (Hg chloride, Sigma, USA) for 30 days. This solution was prepared weekly and changed three times a week. A conversion formula was used to define the tested dose assuming the body area of the animal and its metabolism according to the determined constant (Km) for each species. The constant for humans is Km = 37, while it is Km = 3 for mice [14,56].

The prepared MeHg solution was kept stored in a refrigerator (2–8 °C) protected from light in an opaque plastic bottle. All MeHg manipulation was carried out inside a biosafety cabin, using half-face protection masks and goggles.

After 30 days of MeHg exposure, mice were euthanized with an overdose of ketamine and xylazine. Blood was drawn using heparinized capillaries through the retroorbital plexus. Plasma samples were obtained and stored in a −20 °C freezer. The hippocampi were dissected and immediately snap-frozen in liquid nitrogen, and later stored in a −80 °C freezer until the molecular biology analyses.

### 4.3. Determination of Hair Hg Concentration

Direct analysis of total Hg in hair was carried out by a Direct Mercury Analyzer^®^ (DMA-80 TRICELL, Milestone, Italy). Masses between 1.5 mg and 40.0 mg were weighed and introduced into the automatic sampler under a heating program consisting of a drying temperature of 300 °C (60 s), followed by ashing temperature of 650 °C (210 s) before detection with atomic absorption. The purge time was 60 s, the amalgamator heating time was 12 s, and the signal recording time was 30 s. Oxygen (99.99%; 3.1 bar) was used as the combustion and carrier gas. Several certified reference materials were analyzed during method validation, namely NIST 2709 (San Joaquin Soil), NIST 1641d (Mercury in Water), NIST 2711 (Montana Soil), NIST 1515 (Apple Leaves), and BCR 397 (Trace Elements in Human Hair). The linear range varied between 0.5 and 750.0 ng Hg. The limits of detection and quantification were 0.1 ng Hg and 0.4 ng Hg, respectively, calculated according to the relation between the blank measurements and the angular coefficient of the analytical curve. An accuracy of 92.1% and intermediate precision (in terms of relative standard deviation; n = 7) of 6.6% were achieved for BCR 397 (>90% for all other CRM). All samples were analyzed in triplicate.

### 4.4. Evaluation of Lipid Profile

We measured the plasma total cholesterol and triacylglycerol levels to assess whether MeHg chronic exposure could lead to dyslipidemia. Blood samples were harvested in capillaries and centrifuged at 3500× *g* rpm for 10 min. The total cholesterol and triacylglycerol levels were determined by a semi-automatic analyzer (LabQuest, Labtest, Brazil) using diagnostic kits (Labtest, Brazil).

### 4.5. Systemic Inflammation Biomarkers

For the TNF assay, plasma samples were thawed and homogenized in 1× concentrated phosphate buffer saline. The reagent was diluted in 1% BSA solution, and antibodies and standards were diluted in 1x filtered PBS. After 96-well plate sensitization, the blocking step for nonspecific binding was performed. The plates were incubated overnight, the antibody was added the next day, and the colorimetric reagents were read in a spectrophotometer at 450 nm. Plasma samples were also chemically analyzed for endotoxin concentration (LPS) using a Thermo Scientific Pierce LAL chromogenic endotoxin quantitation kit (Thermo Scientific, Braunschweig, Germany).

### 4.6. Enzymatic Activity of MPO

Hippocampal samples were homogenized in a solution of HTAB (0.5% diluted in sodium phosphate buffer). A total of 100 µL of the supernatant was mixed with 20% acetic acid and 0.5% thiobarbituric acid (diluted in 20% acetic acid, pH 2.4–2.6). The mixture was transferred to a water bath at a temperature of 95 °C for 1 h, sequentially to an ice bath for 30 min, immersed in 8.1% SDS, and immediately centrifuged at 12,000× *g* rpm for 15 min at room temperature. The sample reading was performed in a spectrophotometer under 532 nm. The standard curve was obtained using 1,1,3,3- tetramethoxypropane. The results are expressed in nanomoles of MDA per milligram of tissue (nmol/mg tissue).

### 4.7. Hippocampal MDA Levels

To evaluate cell lipid peroxidation and oxidative stress, the hippocampal samples were homogenized in HTAB solution (0.5% diluted in sodium phosphate buffer (Agar, 1999). The 100 µL aliquots of the supernatant were mixed with 20% acetic acid and 0.5% 2-thiobarbituric acid (diluted in 20% acetic acid, pH 2.4–2.6). The mixture was transferred to a water bath at a temperature of 95 °C, where it remained for 1 h, and subsequently to the ice bath for 30 min. The mixture received 8.1% SDS and was immediately centrifuged at 12,000× *g* rpm for 15 min at 25 °C. The sample reading was performed in a spectrophotometer at 532 nm. The standard curve was obtained using 1,1,3,3-tetra methoxy propane as a standard. The results were expressed in µmol of MDA/mg tissue.

### 4.8. Analysis of Neuroinflammatory-Related Gene Expression

An aliquot of approximately 20 mg of hippocampus was homogenized in RLT buffer using a commercial kit (RNeasy Mini Kit, Qiagen^®^, Hilden, Germany). After maceration, the total RNA was extracted according to the manufacturer’s instructions. The total RNA obtained was quantified in a Nanodrop spectrophotometer (Nanodrop 2000, Thermo Scientific^®^). To obtain the cDNA, a total of 2 μg RNA was standardized. Oligo primers (dT), dNTPs, and the Super Script III enzyme were used according to the manufacturer’s instructions (Invitrogen ^®^, Waltham, MA, USA).

IFN-γ, IL-4, IL-5, IL-6, and IL-10 primers were used. Primers were combined in a reaction mix to produce amplification. Each reaction mix included SYBR Green Master Mix (10 μL), forward and reverse primers (1:1 ratio), nuclease-free water, and cDNA. The complete primers list is depicted in Table 1. Negative controls included only nuclease-free water. The PCR amplification protocol was as follows: AmpliTaq Gold DNA Polymerase starter-activation at 95 °C for 10 min, followed by 40 cycles with denaturation at 95 °C for 15 s, annealing at 62 °C for the 30 s, and extension at 62 °C for 30 s. Amplification was performed on a LightCycler^®^ Nano in a 32-well Real-Time PCR System. Results were expressed as the quantity of amplified gene compared to the control gene β-actin.

### 4.9. Hippocampal Acetylcholinesterase (AChE) Activity

The hippocampal tissues were homogenized in 10% phosphate-buffered saline at pH 7.4. The homogenate was then centrifuged for 15 min at 10,000× *g* rpm and 4 °C. Using bovine serum albumin (BSA) as a standard, the protein content was measured by the Bradford method. The total protein concentration was adjusted to 1 mg/mL. In a 96-well plate, 34 µL of homogenate (1 mg/mL) and 134 µL of 0.75 mM 5,5′-dithiobis (2-nitrobenzoic acid) acid (DTNB) were added, prepared in phosphate buffer, pH 7.4, and the absorbance was measured at 412 nm for a minute. Soon after this procedure, the microplate was removed from the reader and followed by the addition of 34 µL of 9 mM acetylthiocholine iodide (ATC) to each well. We repeated the reading at 412 nm and measured the rate of ATC hydrolysis for 5 min. The AChE activity was measured in nmol/mg of protein per minute. The first reading was used as a normalizing factor due to the coloring of the DTNB.

### 4.10. Hippocampal Amino Acids Levels by HPLC and BDNF ELISA Assay

The analyses of amino acids (glycine, glutamate, taurine, tyrosine, and gamma-amino butyric acid, GABA) were carried out from the dissected hippocampus using high-performance liquid chromatography equipment (Shimadzu, Kyoto, Japan) and a fluorometric detection method. Briefly, frozen samples were homogenized in 0.1 M perchloric acid and sonicated for 30 s at 25 °C. After sonication, samples were centrifuged at 15,000× *g* rpm for 15 min at 4 °C. The supernatants were removed and filtered through a membrane (Millipore, Billerica, MA, the USA) with a suitable amino acid detection column. Brain-derived neurotrophic factor (BDNF) protein expression was measured using a duo-set ELISA assay (R&D systems, Minneapolis, MN, USA).

### 4.11. Behavioral Assessment

All behavioral assessments were conducted blindly in a quiet room devoted to behavioral studies. All apparatuses were handmade and manufactured according to specialized literature. Behavioral tests were analyzed using the AnyMaze video tracking system (Stoelting Co., Wood Dale, IL, USA)

#### 4.11.1. Open-Field Test

The animals were tested for locomotor activity using an open-field apparatus, a black acrylic chamber (30 × 30 cm) with the floor divided into nine squares of equal areas [11]. The animal was placed in the center of the apparatus and left free to explore the environment for 5 min. Parameters such as the total distance covered, total crossed lines, average speed, stopped time, time in the center of the apparatus, and time on the periphery as well as the behaviors of vertical exploration (rearing) and self-cleaning (grooming) were recorded.

#### 4.11.2. Y-Maze Test

Working memory was assessed through spontaneous switching behavior in a Y-maze. The maze consists of three identical arms (45 × 5 × 4.5 cm) positioned at equal angles. Animals were placed on the end of one arm and allowed to move freely through the maze for an 8-min session. The series of arm entries were recorded, and an arm entry was considered complete when the rats’ hind paws were completely placed on the arm. Alternation was defined as successive entries into the three arms in sets of overlapping triplets.

#### 4.11.3. Barnes Maze

The Barnes maze consists of a circular platform 100 cm in diameter with 20 separate holes spread around the circumference and an exhaust box fitted under one of the holes, which is placed in a standard room with visual cues on the periphery. The sound of a bell was used as an aversive stimulus to propel the animal to escape the platform. After habituation, during which the animals were placed in the maze for an uninterrupted 2-min exploration, the mice underwent training tests. If the animal did not find the escape box during the 2-min exploration, it would be guided to the escape box and left there for 1 min. The mice were submitted to two training trials per day with an interval between trials of approximately 15 min for four consecutive days. The amount of time taken to find the exhaust box (latency), the number of incorrect holes explored (errors), the escape velocity, and the total distance traveled were measured. In the probe test, without the exhaust box present, the time spent in the target quadrant (where the exhaust box was previously located) was measured. The probe test was performed 48 h after the last day of training, and the animals were allowed to explore the maze for 2 min during the training and probe sessions.

### 4.12. Statistical Analysis

The sample size of the total experimental mice was the sum of the biochemical analysis and behavioral tests that were calculated to achieve a minimum of 80% power and 95% confidence. Data were initially checked for normality by the Kolgomorov–Smirnov test. Either the Student’s *t*-test or Mann–Whitney test was used accordingly. To assess the statistical differences over time, the data were analyzed by two-way ANOVA, followed by the Bonferroni post hoc test. Results were expressed as the mean ± standard error of the mean (SEM). A *p* < 0.05 value was considered significant. The GraphPad Prism software, version 5.0 (San Diego, CA, USA), was used to perform all of the statistical analyses.

## 5. Conclusions

Altogether, our findings highlight the chronic MeHg-induced effects that may have long-term mental health consequences in prolonged exposed human populations and increase the risk for neurodegenerative diseases later in life.

## Figures and Tables

**Figure 1 ijms-23-13837-f001:**
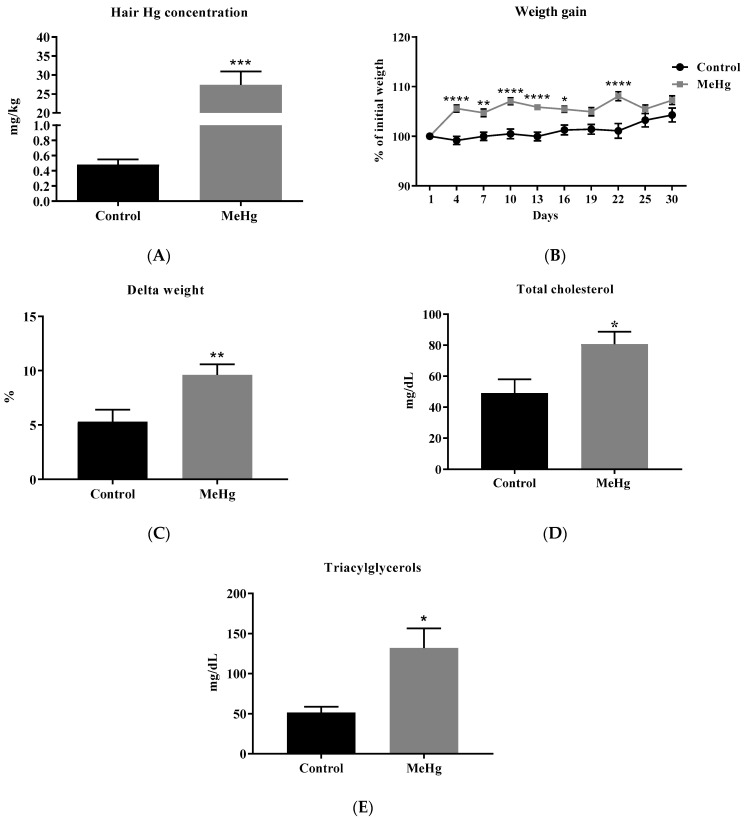
MeHg intoxication and metabolic changes. Values are expressed as the mean ± SEM. (**A**) Hair Hg concentration. *** *p* < 0.001 vs. the control (**B**,**C**). Effect of the oral administration of MeHg (2 mg/L) over 30 days on the body weight percentage of adult C57BL6J mice, n = 20 per group. * *p* < 0.05, ** *p* < 0.001 ****, *p* < 0.0001 vs. control by two-way ANOVA and Bonferroni’s test. (**D**,**E**) Effect of MeHg 30 days of intoxication on the cholesterol and triglyceride plasma levels, n = 6 per group. * *p* < 0.05 vs. control by the unpaired Student *t*-test.

**Figure 2 ijms-23-13837-f002:**
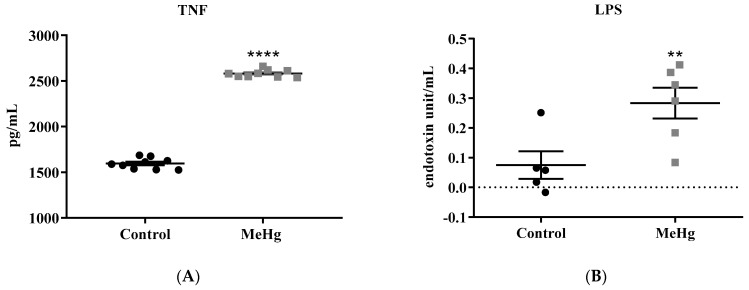
Effect of the oral administration of MeHg (2 mg/L) over 30 days on the systemic inflammation markers. Quantification of plasma TNF (**A**) and LPS (**B**). Values are expressed as mean ± SEM, n = 6 per group. **** *p* < 0.0001, ** *p* = 0.01 vs. control by the unpaired Student *t*-test.

**Figure 3 ijms-23-13837-f003:**
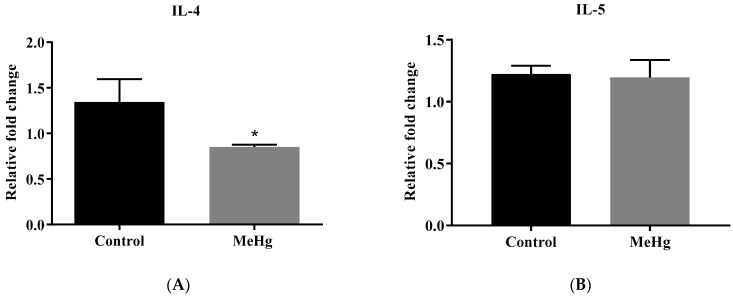
Hippocampal neuroinflammatory-related biomarkers. Values are expressed as mean ± SEM. (**A**–**D**) Neuroinflammatory-related gene expression in the hippocampus from MeHg-challenged mice. n = 6 per group, * *p* < 0.05 vs. control by the unpaired Student *t*-test. (**E**) Effect of the oral administration of MeHg (2 mg/L) over 30 days on the hippocampal MPO. n = 6 per group, * *p* < 0.05 vs. control by the unpaired Student T test (**F**) Effect of MeHg over 30 days on the hippocampal MDA concentration. n = 6 per group, ** *p* < 0.005 vs. control by the unpaired Student *t*-test.

**Figure 4 ijms-23-13837-f004:**
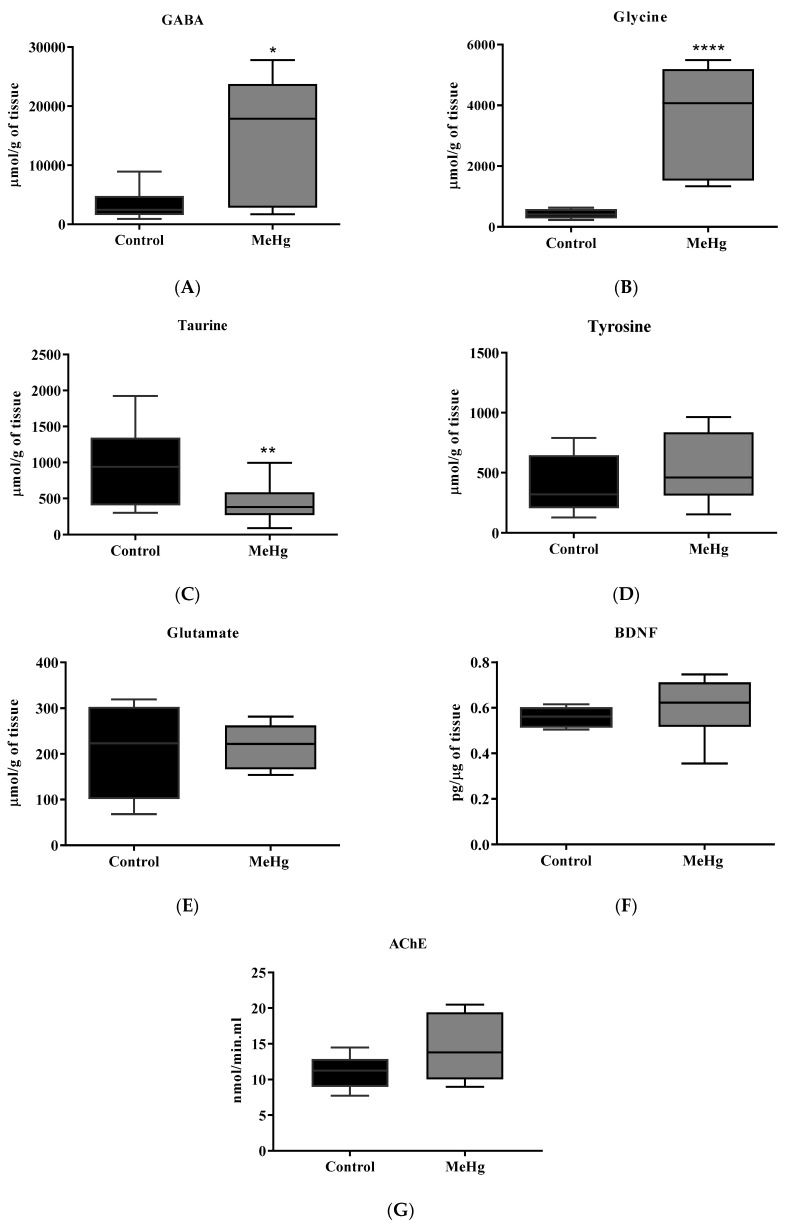
MeHg hippocampal amino acid levels. (**A**–**E**) Effect of oral administration of MeHg (2 mg/L) over 30 days on the hippocampal concentration of GABA, glycine, taurine, glutamate, and tyrosine. Values are expressed as the mean ± SEM, n = 8 per group. * *p* = 0.01, ** *p* = 0.003, **** *p* < 0.0001 by the unpaired Student *t*-test. (**F**) Effect of oral administration of MeHg (2 mg/L) over 30 days on the BDNF hippocampal concentration. Values are expressed as the mean ± SEM, n = 8 per group. (**G**) Effect of the oral administration of MeHg (2 mg/L) over 30 days on the hippocampal acetylcholinesterase (AChE) activity of adult C57BL6J mice. Values are expressed as mean ± SEM, n = 8 per group by the unpaired Student *t*-test.

**Figure 5 ijms-23-13837-f005:**
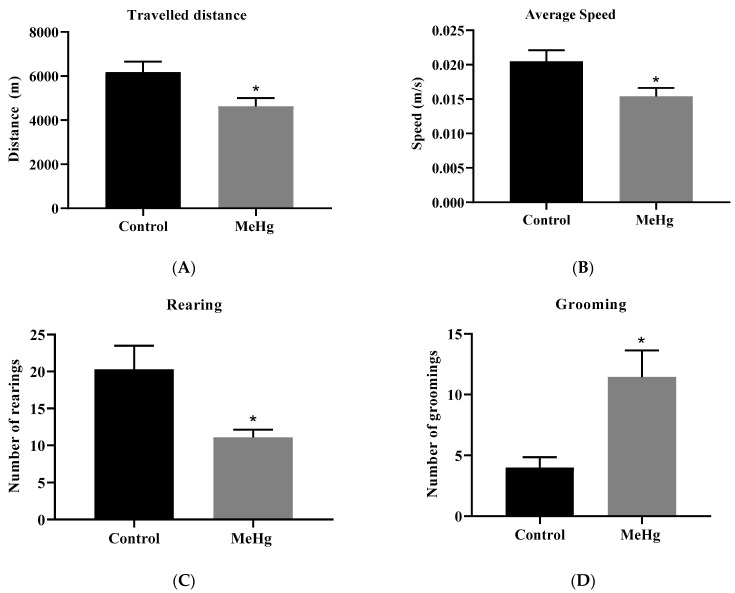
Exploratory and locomotor behavior. Effect of the oral administration of MeHg (2 mg/L) over 30 days, assessing space exploration including travelled distance (**A**) and averaged speed (**B**), rearing (**C**), and also self-grooming (**D**) behaviors in the open field test. Values are expressed as the mean ± SEM, n = 10 per group. * *p* = 0.01 by the unpaired Student *t*-test.

**Figure 6 ijms-23-13837-f006:**
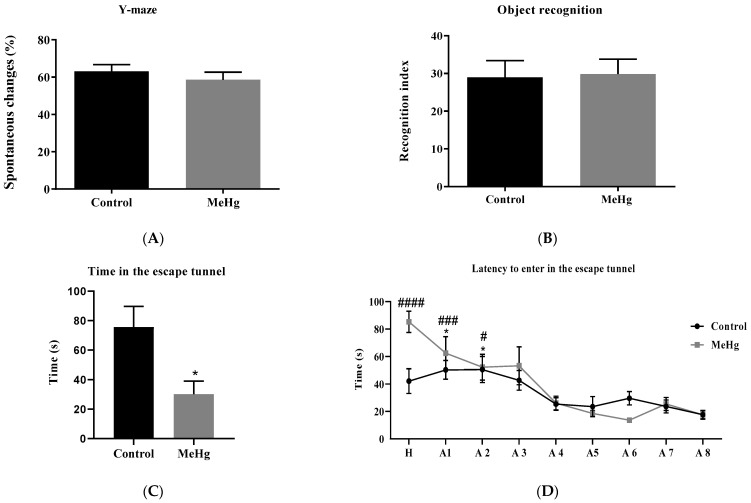
Memory behavior assessment. Effect of oral administration of MeHg (2 mg/L) over 30 days on mouse behavior using (**A**) Y-maze and (**B**) object recognition test. Values are expressed as mean ± SEM, n = 10 per group. unpaired Student *t*-test. Effect of oral administration of MeHg (2 mg/L) over 30 days on mouse memory using the Barnes maze (**C**,**D**). Values are expressed as the mean ± SEM, n = 10 per group. * *p* < 0.05 vs. control (**C**). * *p* < 0.05, vs. A8 control. # *p* < 0.05, ### *p* < 0.001, #### *p* < 0.0001 vs. A8 MeHg by two-way ANOVA (**D**). H = Habituation, A = Acquisition, P = Probe.

**Table 1 ijms-23-13837-t001:** List of the study primers.

Primers	Forward (5′→3′)	Reverse (5′→3′)
IFN-γ	CACGCCGCGTCTTGGT	TCTAGGCTTTCAATGAGT
IL-4	ACAGGAGAAGGGACGCCAT	GAAGCCCTACAGACGAGCTCA
IL-5	GACTCTCAGCTGTGTCTGGG	GGACAGCTGTGTCAAGGTCT
IL-6	CTGCAAGAGACTTCCATCCAG	AGTGGTATAGACAGGTCTGTTGG
IL-10	GTGAAGACTTTCTTTCAAACAAAG	CTGCTCCACTGCCTTGCTCTTATT
β-actin	GTGGGCCGCTCTAGGCACCAA	CTCTTTGATGTCACGCACGATTTC

IFN-γ = interferon gamma; IL = interleukin.

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
