# Peer review of "Chronic Methylmercury Intoxication Induces Systemic Inflammation, Behavioral, and Hippocampal Amino Acid Changes in C57BL6J Adult Mice"

_ijms, 2022, doi:10.3390/ijms232213837_

Round 1

Reviewer 1 Report

The manuscript by Nascimento and co-workers addresses an interesting and pertinent topic, since it is relevant to study methylmercury toxicological effects to better plan preventive measures, elaborate public health policies and optimize treatment strategies.

This manuscript reports a well-reasoned animal study, combining behavioral, biochemical and molecular biology approaches. The results have potential applicability in the clinical settings, for which it represents an added value. The draft is well structured. The Discussion integrates and correlates the results and appropriately recognizes study limitations and future work opportunities.

Despite its current flaws, which I detail below, I believe that this paper fits the scope of the IJMS and that it could be considered for publication, once the following concerns are met.

General remarks

- I recommend that the manuscript is revised for formatting, since some aspects (e.g., subscripts and superscripts) still deserve corrections. Although I have specified some (see below), I may not have mentioned all.

- According to the instructions for authors, acronyms/abbreviations/initialisms should be defined the first time they appear in each of three sections: the abstract; the main text; the first figure or table. When defined for the first time, the acronym/abbreviation/initialism should be added in parentheses after the written-out form. Please make sure this is done for all acronyms/abbreviations/initialisms (e.g., GABA, LTP, TNF, BHA, etc.).

Abstract

- Line 32: Please replace “undergone” by “underwent”.

Introduction

- Line 51: Please replace “prolonged” by “prolongedly”.

- Why did the study focus on the hippocampus, and not on other brain structures? Please provide a brief rationale on this, in order to contextualize the reader about this choice.

Results

- Line 145: Please replace “1ª and 2ª acquisition tests” by “1st and 2nd acquisition tests”.

Discussion

- Lines 190-191, 196, 203 and 206: Please write the “2” in “H2O2”, “NO2” and “O2” as a subscript.

- Line 201: Please define “Nrf2” in full, before providing its abbreviation.

- Line 254: Please replace “1ª and 2ª acquisition” by “1st and 2nd acquisition”.

- Lines 255-256: Please write the “-1” in “kg-1 day-1” as a superscript.

Materials and Methods:

- Section 4.1: How was the total number of animals in the study defined? Please add some remarks on sample size.

- Section 4.1: Please specify the number of animals per group. One may assume that each group is composed of 60 animals, but that is not explicitly stated.

- Line 344: Please replace “store” by “stored”.

- Line 415, table 1: Please specify whether the primer sequences are given from 5’ to 3’.

- Section 4.11: Were the apparatuses used for behavioral assays handmade or purchased from a commercial company? Please specify. Also, please detail how the multiple behavioral parameters were recorded.

- Line 447: Please add “were recorded” (or an analogous expression) at the end of the sentence.

- Lines 478-480: Please delete the guidelines of the IJMS template regarding the “Conclusions” section or replace them by an actual section.

Author Response

We are grateful for your kind consideration. The manuscript was revised for formatting and all requested changes. About the hippocampus as a brain structure of choice, we consider the fact that the hippocampus is a known brain area affected by Alzheimer's disease, as MeHg has been shown to contribute and being a risk factor augmented risk of dementia in vulnerable populations, we choose it as a focus in this study. Regarding the sample number, each group is composed of 60 animals; this value summarizes the sum of the biochemical analysis and behavioral tests sample number that was calculated to achieve a minimum of 80% power and 95% confidence in the Statistical analysis tests. Regarding the behaviors, all the apparatuses were handmade and manufactured according to specialized literature. Behavioral tests were analyzed using the AnyMaze video tracking system (Stoelting Co., Illinois, USA).

Reviewer 2 Report

This manuscript investigates the induction of inflammation, fluctuations in hippocampal amino acids, and behavioral abnormalities in a number of mice exposed to MeHg in drinking water. Clear data are obtained, but I am curious to comment on the following.

1.      In general, I think that MeHg induces a decrease in body weight in mice, but the mice in this experiment, on the contrary, showed a significant increase in body weight. The authors have discussed the involvement of thyroid hormone dysregulation by MeHg, but have you actually measured plasma free thyroxine/triiodothyronine ratio?

2.      This weight gain was observed early in MeHg administration and not after 25 days of MeHg administration. Can you give me any discussion on this?

3.      In this MeHg administration experiment, what was the degree of accumulation in the liver and its toxicity?

4.      Please correct the illegible "µ" on the vertical axis of figures.

5.      In 2.6 of "Results," Figure 6C and Figure 6D are reversed.

Author Response

We are grateful for your kind consideration. Regarding the weight gain revised questions, other previous works showed similar results. Ferrer and collaborators (2021) showed that Chronic exposure to MeHg induces body weight gain due to ghrelin-related hypothalamic AMP-activated kinase (AMPK)/uncoupled protein 2 (UCP2) signaling since ghrelin is a known orexic gastric hormone that increases appetite, the increase in food consumption and consequent increase in weight gain is expected. MeHg-induced weight gain in our study reaches a platot in the experiment, suggesting other metabolic adjustments. The intestinal microbiota and environmental conditions may also influence MeHg effects on weight. However, more studies are needed to dissect the fine regulation of body growth, fat deposits, and weight gain due to MeHg intoxication in adult mice. Regarding liver toxicity, previous works of our group showed that MeHg induces Hg accumulation in adult mice's livers. Others works showed that this accumulation leads to hepatic dysfunction with increased TBARS and serum levels of transaminases, effects that may partly explain dyslipidemia seen in our study.
